# Identifying robust predictors of treatment response in trauma-affected refugees: Results from a randomised controlled trial

Jessica Carlsson[1,2]*, Maja Bruhn[1,2], Ida Sophie Poschmann[1], Derrick Silove[1,3], Maria Lurenda Westergaard[1], Erik Lykke Mortensen[4], Hinuga Sandahl[1]

1 Mental Health Center Ballerup, Copenhagen University Hospital - Mental Health Services CPH, Ballerup, Denmark, 2 Department of Clinical Medicine University of Copenhagen, Copenhagen, Denmark, 3 School of Psychiatry, University of New South Wales, Sydney, New South Wales, Australia, 4 Department of Public Health, University of Copenhagen, Copenhagen, Denmark

* jessica.carlsson.lohmann@regionh.dk

## Abstract

### Background

Given the high prevalence of trauma-related mental health problems in a growing population of refugees, identifying robust predictors of treatment outcome can optimize treatment strategies and resource allocation. The current study aimed to examine the robustness of predictors from seven previous predictor studies at the Competence Centre for Transcultural Psychiatry (CTP).

### Methods

We studied if predictors from previous studies could be replicated in a sample of 219 refugees with PTSD who participated in a new distinct study conducted at CTP. We included all variables previously found to be significantly related to treatment response, as measured via the Harvard Trauma Questionnaire (HTQ) and the Hamilton Depression Scale (HAM-D). We categorised the variables as sociodemographic factors, CTP predictor index items, and baseline assessment scores. We performed regression analyses with one exposure variable at a time, controlling for baseline assessment scores corresponding to the dependent variable. We also conducted multiple linear regression analyses to examine the variance explained in total by the identified significant predictor variables.

### Results

The study successfully replicated several predictors of treatment response for PTSD and depression, including a high understanding of therapy concepts, acceptability of psychological treatment, ability to reflect, motivation for active participation and cognitive resources, as well as low baseline pain levels. The findings replicated a

**Data availability statement:** The data that support the findings of this study are available upon request from the Competence Centre for Transcultural Psychiatry www.ctp-net.dk. The data are not publicly available due to privacy and ethical restrictions. Public deposition would breach compliance with the protocol approved by the Ethics Committee of the Capital Region of Denmark and the Danish Data Protection Agency. Furthermore, public availability would compromise patient privacy.

**Funding:** The RCT was supported by grants from TrygFonden (JC, grant number 120354) and Fonden til Lægevidenskabens Fremme (HS, grant number 16-319). There was no additional external funding received for this study. All funders did not play a role in the study design, data collection, analysis, decision to publish, or preparation of the manuscript.

**Competing interests:** The authors have declared that no competing interests exist.

U-shaped relationship between age and treatment response for depression, indicating better outcomes for younger participants and those over 60. Multiple regression analyses explained 28% and 39% of the variance in the PTSD (HTQ) and depression (HAM-D) treatment responses, respectively.

## Conclusions

This study contributes to the understanding of treatment response predictors in trauma-affected refugees by replicating findings from previous research in a new sample. It highlights the importance of modifiable factors, such as psychotherapy readiness, and underscores the necessity for tailored interventions to enhance treatment efficacy. As the global refugee crisis grows, the insights from this research can inform mental health strategies, ultimately improving care for this vulnerable population. Future research should continue to investigate modifiable predictors to further enhance treatment outcomes.

## Trial registration

ClinicalTrials.gov NCT02761161

## Introduction

Identifying predictors of treatment response is an important focus of research aimed at developing better treatment options. With an increasing number of trauma-affected refugees, this avenue of research is called for due to a pressing need to effectively utilise scarce resources for this vulnerable population. As of mid-2024, the number of refugees reached 37.8 million; compared with 14.4 million refugees 10 years ago [1,2]. Trauma-affected refugees are at high risk of developing mental health problems, with an estimated prevalence of 30% for PTSD and depression [3,4]. Both pre- and postmigration factors contribute to these mental health problems [5–7]. Psychosocial interventions for trauma-affected refugees have resulted in small- to medium-sized improvements [8–10] and most often need to be comprehensive and multidisciplinary to address the complex health conditions and challenges of trauma-affected refugees [9–11]. Hence, the growing number of refugees, combined with the costly nature of mental health interventions and lack of improvement during these interventions, emphasise the need to optimise treatment strategies to improve treatment response and make more effective use of resources [12].

### Need for studies on predictors of treatment response

Importantly, robust predictors of treatment outcome can assist healthcare personnel in assigning interventions that are more likely to be successful, thereby optimising the use of resources.

Efforts have been made to determine which factors might predict treatment response in trauma-affected refugees, given that not all individuals show a positive

response to best-practice treatments. In previous reseach, several possible predictors of treatment response have been identified; including sociodemographic factors; exposure to traumatic events; and factors directly related to mental and physical health, such as the severity of symptoms and comorbidities [12–15].

Past studies examining predictors of treatment response are nevertheless limited by small sample sizes, differences in the factors considered as possible predictors, and variations in outcome measures, all of which contribute to inconsistent findings [13,14,16–20]. Thus, although several predictors of treatment response have been identified, knowledge on robust predictors of treatment response in psychosocial interventions remains limited. Studies reproducing previous findings are necessary but nevertheless lacking [21,22].

The identification of modifiable predictors in treatment, such as sleep or motivation, presents an opportunity for providing treatment interventions aimed specifically at improving these areas, thereby increasing the likelihood of success of the intervention [23]. Conversely, identifying non-modifiable factors such as offender status or age can provide insights into the necessity of early identification and may assist in assigning interventions that are more likely to be successful [24]. While such factors cannot be changed, they can still inform clinical decision-making; for example, individuals experiencing persistent guilt or rumination may benefit from trauma-focused interventions that address these manifestations of distress.

### Replicating findings from previous predictor studies

At the Competence Center for Transcultural Psychiatry (CTP), a specialised mental health outpatient clinic in Denmark, we have performed a total of five randomised controlled trials (RCTs) with trauma-affected refugees. On the basis of data from the first four trials, we have previously published seven papers identifying different predictors of treatment response [12,24–29]. However, we have not examined whether the predictors of treatment response identified across these four trials can be replicated in a similar but new and distinct sample from our fifth randomized controlled trial [30,31]. Predictors from the first four trials are summarised in Table 1.

The standard treatment in all five RCTs was a multidisciplinary psychosocial intervention. Sociodemographic and mental health data for the populations included in the five RCTs show only unsystematic differences, allowing for comparisons between studies [32–36].

The aim of the current study was to explore whether the findings of treatment outcome predictors from previous CTP studies are robust and can be replicated in this study's population. The current study examines the robustness of these factors by repeating the analysis with the identified possible predictors in a new sample of 219 trauma-affected refugees who participated in an RCT conducted at CTP. Furthermore, we examine the variance explained in total by the identified significant predictor variables.

Improved knowledge on factors influencing treatment response will enable offering improved treatment.

## Materials and methods

### Study design

Data for the current study derived from a a randomized controlled superiority trial with an allocation ratio of 1:1:1:1 and an allocation sequence with block size unknown to the investigator. The randomization was stratified by gender.

The patients were randomised into four groups. All four groups received treatment as usual (TAU), which consisted of a 10-month intervention that included manual psychotherapy, physiotherapy, psychopharmacotherapy, psychoeducation, and social counselling. One group received solely TAU, serving as a control group; the three remaining groups were active treatment groups receiving add-on treatment with either mianserin, IRT, or a combination of both.

The primary outcome in the RCT was the *Pittsburgh Sleep Quality Index*. Based on previous studies, a minimal clinically important difference of 2.5 points and a standard deviation of 3 were assumed. With 90% power and alpha = 0.05, the required sample size was 128. To account for an expected dropout of 25%, the target sample was increased to 228 participants.

Table 1. Summary of predictors of positive treatment response identified in four RCTs.

| | RCT-1 (n = 85) Buhmann et al. 2015 | RCT-3 (n = 195) Sonne et al. 2016 | RCT-4 (n = 318) Nordbrandt et al. 2022 Nordbrandt (thesis) | RCT-2 and 3[1] (n = 321) Sonne et al. 2021 | RCT-1–4[2] Sander et al. 2019 (n = 825) Westergaard et al. 2023 (n = 892) |
|---|---|---|---|---|---|
| **Socio-demographic factors** | | | | | |
| Age | | | HTQ | HTQ | |
| Few years since arrival in Denmark | | | | HTQ | |
| Migrant status: Refugee | | | | HTQ | |
| No combat experience | | | | HAM-D | |
| Stable occupation/ not being on public financial support | HTQ | | | | |
| Good Danish language skills/ no need for interpreter | | | | | HTQ |
| Shorter duration of functional impairment | | | HTQ | | |
| **CTP predictor index** | | | | | |
| Safe and good upbringing | | HAM-D | | | |
| Haven't received previous psychiatric treatment | | HAM-D | HTQ | | |
| No chronic pain | | HAM-D | HTQ | | |
| No chronicity of mental condition | | | HTQ | | |
| Good understanding of the concept of psychotherapy | | HAM-D | HTQ | | |
| High acceptability to psychological treatment | | HAM-D | HTQ | | |
| Good ability to reflect on own circumstances | | HAM-D | HTQ | | |
| High motivation for active participation | | HAM-D | HTQ | | |
| High cognitive resources | | HAM-D | HTQ | | |
| Full time employment | | HTQ | | | |
| Higher education | | | HTQ | | |
| **Baseline assessment scores** | | | | | |
| High GAF-F | | | | HTQ HAM-D | |
| Low pain score, BPI | | | HTQ | | |
| No severe headache (HSCL item 8) | | | | | HTQ |

[1]Pooled data from RCT 2 and 3. [2]Pooled data from RCT 1, 2, 3, and 4.

HTQ: Harvard Trauma Questionnaire. HAM-D: Hamilton Rating Scale Depression, GAF-F: Global Assessment of Functioning – function. BPI: Brief Pain Inventory, HSCL: Hopkins Symptom Checklist-25.

Please see study protocol and paper reporting main results for in-depth description of methods and results of the RCT [30,36]. The first patient was enrolled on the 12th of May 2016 the primary completion date was the 1st of June 2019.

### Ethics approval and consent to participate

All the participants provided signed informed consent. The trial was approved by the Danish Medicines Agency, the Ethics Committee of the Capital Region of Denmark, and the Danish Data Protection Agency. The study was registered at ClinicalTrials.gov, ID: NCT02761161, on 27 April 2016 (https://clinicaltrials.gov/study/NCT02761161?id=NCT02761161&rank=1). The authors assert that all procedures contributing to this work comply with the ethical standards of the relevant

national and institutional committees on human experimentation and with the Helsinki Declaration of 1975, as revised in 2008.

## Participants

The study sample consisted of 219 trauma-affected refugees, the intention-to-treat sample of the RCT. Given that there were no differences in outcomes across the intervention groups [36], the current study examined the complete sample across intervention groups. The participants were adult refugees (or persons whose families reunified with refugees) diagnosed with PTSD according to the ICD-10.

## Measures

**Variable selection.** *Outcome measures*: For the present study, PTSD symptoms were measured via self-reports on the Harvard Trauma Questionnaire (HTQ) [37,38], and symptoms of depression were measured via observer ratings on the Hamilton Depression Scale (HAM-D) [39].

*Exposure variables*: We selected candidate predictor variables on the basis of an assessment of the seven previous CTP predictor studies [12,24–28,40]. We included all variables previously found to be significantly related to the HTQ or HAM-D via correlational, bivariate, or multiple regression analyses [12,24–28,40]. We categorised the variables as socio-demographic factors, CTP predictor index items, and baseline assessment scores. The variables are presented in Table 1.

The CTP predictor index (S1 Appendix) was developed in 2012 to assess predictors of treatment response on the basis of clinical experience and available literature on trauma-affected refugees [12,41]. The index initially consists of 15 items rated on a 0–4 Likert scale (4 being the best score) according to predefined criteria [12]. A medical doctor rated five items, a psychologist rated five items, and a social counsellor rated five items. The index was later modified on the basis of experiences from previous CTP predictor index studies and the needs of ongoing RCTs [26,27,30,42].

The following candidate variables were included in the analyses:

1. Sociodemographic factors included were *age, years since arrival in Denmark, gender, migrant status refugee, combat experience, need for an interpreter in psychologist sessions, education above ten years in the country of origin, stable occupation/not on financial support, and duration of functional impairment.*

2. *The* CTP predictor index items included were *upbringing, previous psychiatric treatment attempts, chronic pain, chronicity of the mental condition, understanding of concepts of therapy, acceptability of psychological treatment, ability to reflect, motivation for active participation, cognitive resources, daily activity, employment status, and economy (self-perceived).*

3. Baseline assessment scores included were *Global Assessment of Functioning (GAF-F)* [43], the *Brief Pain Inventory-Short Form (BPI)* [44], and *headache* measured on item 8 of the HSCL-25.

## Statistics

To describe the characteristics of the participants, means and standard deviations were calculated for continuous variables, and percentages were calculated for categorical variables. The mean rating scores were calculated pre- and post-treatment, and pre- and posttreatment differences were tested with paired t-tests.

We tested our main hypothesis that previous treatment predictors could be replicated in this study population by performing regression analyses with one exposure variable at a time, controlling for baseline assessment scores corresponding to the dependent variable. In previous predictor studies, correlation analyses, bivariate regression analyses, and hierarchical regression analyses were performed, and different variables were controlled for in the hierarchical regression analyses. In the present study, we chose to perform regression analyses that only controlled for baseline scores to compare variables across studies.

 

We used pre- to posttreatment score differences in symptoms of PTSD (HTQ) and depression (HAM-D) as dependent variables and included the variables mentioned above as exposure variables.

To supplement the regression analyses, we performed multiple linear regression analyses in two models to examine the variance explained in total by the identified significant predictor variables. In Model 1, we included all the sociodemographic factors, baseline assessment scores identified as significant in the regression analyses, and the baseline assessment scores corresponding to the dependent variable to adjust for baseline symptoms. In Model 2, CTP predictor index items identified as significant in regression analyses were added to the model. Based on the results of the initial multiple linear regression analyses, we tested an alternative (Model 2b) regression analysis to avoid the effects of possible collinearity among the five CTP predictor index items rated by a psychologist. Instead of the individual predictor items, this analysis included the first principal component derived from these five items.

In the initial analyses, we included an indicator of the intervention group in the RCT. However, we did not identify significant differences between the intervention groups, and the variable was nonsignificant in the regression analyses. Hence, the presented analyses do not include an indicator of the intervention group.

To assess assumptions of linearity, a scatterplot of the included continuous variables was plotted with a superimposed regression line. Visual inspection of these plots revealed a linear relationship between the variables and no outliers of importance. For the variable age, the relationship with treatment response was nonlinear for several dependent variables; thus, both linear and quadratic terms for age were included.

Assumptions of independence of errors, homoscedasticity, and normality of residuals were met to a reasonable degree based on visual inspection of residual-versus-fitted plots and Q-Q plots, as well as rvfplot commands in Stata. Multicollinearity was assessed using variance inflation factors (VIF), and no problematic levels were observed.

All analyses were conducted via STATA 17. Linear regression analyses were performed via STATA's SEM procedure and Full Information Maximum Likelihood analyses to include all available information, including data for participants with missing posttreatment outcome scores.

## Results

The study included 219 participants. The majority came from Syria, Afghanistan, Iraq, and Iran. Trauma histories were diverse: 98% had experienced war, 42% had been imprisoned, and 37% had experienced violence from relatives. Further characteristics and the included exposure variables are presented in Table 2.

The mean scores before and after treatment and the changes from pre- to posttreatment for the HTQ and HAM-D are presented in Table 3. We found significant improvement in scores on the HTQ ($p = 0.000$) but not on the HAM-D ($p = 0.6316$).

Table 4 presents the results of regression analyses between candidate predictors and treatment response for symptoms of PTSD and depression controlled for baseline values in the current sample.

### Replicating findings from previous predictor studies

We identified the following significant variables as predictors of greater symptom improvement in symptoms of PTSD (HTQ): younger age ($p = 0.017$), migrant status (refugee) ($p = 0.034$), greater understanding of the concepts of therapy ($p = 0.000$), higher acceptability of psychological treatment ($p = 0.000$), stronger ability to reflect ($p = 0.000$), higher motivation for active participation ($p = 0.017$), high cognitive resources ($p = 0.001$), and low pain score on BPI at baseline ($p = 0.001$). The variable with the greatest association was *understanding the concepts of therapy* ($\beta = 0.32$).

For depression (HAM-D), the following variables significantly predicted greater symptom improvement: greater understanding of the concepts of therapy ($p = 0.016$), higher acceptability of psychological treatment ($p = 0.003$), stronger ability to reflect ($p = 0.001$), higher motivation for active participation ($p = 0.008$), high cognitive resources ($p = 0.007$), absence of chronic pain ($p = 0.034$) and low pain score on BPI at baseline ($p = 0.034$). In contrast, medium duration of functional

**Table 2. Characteristics of participants (N = 219).**

| Sociodemographic data | | Mean | (SD) |
|---|---|---|---|
| Age | | 44.44 | (10.40) |
| Years since arrival in Denmark | | 13.34 | (9.55) |
| | | **N** | **(%)** |
| Female gender (*n* = 219) | | 110 | (50.00) |
| Migrant status (*n* = 218)*: | Refugee | 165 | (75.69) |
| | Family reunified | 53 | (24.31) |
| Combat experience (*n* = 189)* | | 47 | (24.87) |
| Education < 10 years in country of origin (*n* = 165)* | | 79 | (47.88) |
| Stable occupation/not on financial support (*n* = 196)* | | 15 | (7.65) |
| Need for interpreter in psychologist sessions (*n* = 140)* | | 95 | (67.86) |
| Functional impairment: (*n* = 175)*: | < 5 years | 128 | (73.14) |
| | 5-10 years | 23 | (13.14) |
| | > 10 years | 24 | (13.71) |
| **CTP predictor index items** | | **Mean** | **(SD)** |
| *Rated by a medical doctor* | | | |
| Upbringing | | 2.01 | (1.23) |
| Previous psychiatric treatment attempts | | 2.19 | (1.18) |
| Chronic pain | | 1.99 | (1.14) |
| Chronicity of mental condition | | 3.33 | (0.75) |
| Mean score items rated by a medical doctor | | 1.93 | (0.69) |
| *Rated by a psychologist* | | | |
| Understanding the concepts of psychotherapy | | 1.98 | (1.14) |
| Acceptability of psychological treatment | | 2.61 | (1.02) |
| Ability to reflect | | 2.35 | (1.00) |
| Motivation for active participation | | 2.34 | (1.10) |
| Cognitive resources | | 2.23 | (1.00) |
| Mean score items rated by a psychologist | | 2.30 | (0.87) |
| *Rated by a social counsellor* | | | |
| Social relations | | 2.30 | (0.94) |
| Daily activity | | 1.61 | (1.32) |
| Employment status | | 1.78 | (0.89) |
| Economy (self-perceived) | | 1.97 | (1.27) |
| Integration | | 2.50 | (1.06) |
| Mean score items rated by a social counsellor | | 2.22 | (0.65) |
| **Baseline assessment scores** | | **Mean** | **(SD)** |
| GAF-F baseline | | 51.57 | (7.86) |
| BPI baseline | | 6.34 | (1.96) |
| Headache (HSCL item 8) baseline | | 3.13 | (0.83) |

SD: Standard Deviation. GAF-F: Global Assessment of Functioning – Function. BPI: Brief Pain Inventory. HSCL: Hopkins Symptom Checklist-25. N: 140–219 (due to missing).

*Data not available for all randomized participants.

**Table 3. Mean score pre- and post-treatment, difference pre-post treatment and p-value for HTQ and HAM-D.**

| Rating | Mean pre-treatment score (SD) | Mean post-treatment score (SD) | Mean difference pre-post treatment (SD) | p-value |
|---|---|---|---|---|
| **HTQ** | 3.14 (0.41) | 2.95 (0.62) | 0.18 (0.59) | 0.0001 |
| **HAM-D** | 22.28 (5.49) | 21.97 (7.99) | 0.31 (7.71) | 0.6316 |

Difference scores and significance according to paired t-test.

SD: Standard Deviation.

HTQ: Harvard Trauma Questionnaire. HAM-D: Hamilton Rating Scale Depression.

N: Range 145–159 (due to missing values post-treatment).

**Table 4. Bivariate regression results controlled for baseline value.**

| | HTQ change pre-post | | HAM-D change pre-post | |
|---|---|---|---|---|
| **Sociodemographic data** | β | p-value for β | β | p-value for β |
| Age (c) | −0.18 | 0.017* | −0.18 | 0.058 |
| Gender (b) | 0.03 | 0.720 | 0.01 | 0.886 |
| Years since arrival in Denmark (c) | −0.03 | 0.661 | −0.15 | 0.067 |
| Migrant status refugee (b) | −0.15 | 0.034* | −0.11 | 0.151 |
| Combat experience (b) | 0.09 | 0.218 | 0.04 | 0.610 |
| Education < 10 years in country of origin (b) | −0.16 | 0.067 | −0.03 | 0.671 |
| Stable occupation/not on financial support (b) | −0.02 | 0.829 | −0.03 | 0.671 |
| Need for interpreter in psychologist sessions (b) | 0.13 | 0.125 | 0.01 | 0.919 |
| Duration of functional impairment: | | | | |
| Medium | -0.10 | 0.097 | -0.19 | 0.005* |
| Long | 0.01 | 0.890 | 0.02 | 0.674 |
| **CTP predictor index items (c)** | | | | |
| Upbringing | 0.09 | 0.256 | −0.02 | 0.803 |
| Previous psychiatric treatment attempts | 0.02 | 0.787 | −0.09 | 0.270 |
| Chronic pain | 0.09 | 0.259 | 0.19 | 0.034 |
| Chronicity of mental condition | 0.13 | 0.179 | 0.00 | 0.975 |
| Understanding the concepts of psychotherapy | 0.32 | 0.000*** | 0.21 | 0.016* |
| Acceptability of psychological treatment | 0.28 | 0.000*** | 0.23 | 0.003* |
| Ability to reflect | 0.28 | 0.000*** | 0.27 | 0.001*** |
| Motivation for active participation | 0.27 | 0.017* | 0.24 | 0.008* |
| Cognitive resources | 0.26 | 0.001*** | 0.24 | 0.007** |
| Daily activity | 0.08 | 0.362 | 0.05 | 0.567 |
| Employment status | 0.09 | 0.314 | 0.03 | 0.671 |
| Economy (self-perceived) | 0.01 | 0.909 | −0.02 | 0.806 |
| **Rating scales (c)** | | | | |
| GAF-F baseline | −0.03 | 0.675 | 0.00 | 0.970 |
| BPI baseline | −0.26 | 0.001*** | −0.19 | 0.034* |
| Headache (HSCL item 8) baseline | −0.13 | 0.084 | −0.05 | 0.574 |

HTQ: Harvard Trauma Questionnaire. HAM-D: Hamilton Rating Scale Depressiony. HSCL: Hopkins Symptom Checklist-25. GAF-F: Global Assessment of Functioning – function. BPI: Brief Pain Inventory.

b: bivariate (1=yes, 2=no). c=continuous.

impairment was associated with less symptom improvement (i.e., poorer treatment outcome) (p = 0.005). The variable with the strongest association was the *ability to reflect* (β = 0.27).

## Variance explained by the identified significant predictor variables

Table 5 presents the results of the multiple linear regression analyses and shows the variance explained in total by the identified significant predictor variables. Model 1 included all the sociodemographic factors and baseline assessment scores identified as significant in the regression analyses and the baseline assessment scores corresponding to the dependent variable, to adjust for baseline symptoms. Model 1 accounted for 19% and 32% of the variance in improvement in symptoms for PTSD and depression, respectively.

In Model 2, the CTP predictor index items found to be significant in the regression analyses were added to Model 1, increasing the explained variance by 9% and 5%. The model now accounted for 28% and 39% of the variance in symptom improvement for PTSD and depression, respectively. In Model 2, age added significantly to the prediction of depression symptom improvement (HAM-D) (p = 0.000). Furthermore, the quadratic term included for age was significantly associated with treatment response for depressive symptoms (HAM-D) (p = 0.000), reflecting a nonlinear relationship. For HAM-D, the association between age and treatment response showed a U-shaped relationship, indicating a greater treatment response, primarily for participants younger than 40 years but also for the oldest group above 60 years.

In Model 2, which included all the CTP predictor index variables, none of these variables were significantly associated with any of the dependent variables. Given the significance observed in analyses of the five individual CTP predictor items rated by a psychologist, the lack of significance in Model 2 might reflect collinearity among the predictor variables. Consequently, a principal component analysis was conducted on the five predictor variables. Only one eigenvalue was greater than 1, and the first principal component explained 69% of the variance. All the variables had high loadings on the first principal component, ranging from 0.77 (cognitive resources) to 0.88 (ability to reflect). When scores on the first principal component were included in alternative Model 2b together with the variables from Model 2, the component was a highly significant predictor of symptom improvement for both PTSD (HTQ) (β = 0.30 p = 0.000) and depression (HAM-D) (β = 0.25 p = 0.001).

Furthermore, owing to the similarly high loadings on the first principal component for the five CTP predictor items rated by a psychologist, we calculated the mean score for the five items. The correlation coefficient between the mean score and the first principal component was close to 1, and essentially the same results were observed when the mean score replaced the first principal component in a regression model.

## Discussion

Repeated studies have shown only small- to medium-sized improvements in the treatment of trauma-affected refugees [8–10]. Thus, we still need to explore how we can improve the support offered to this group. With several possible predictors of treatment response identified in previous studies in the same setting, the current study examined the robustness of these factors by repeating the analysis with the identified possible predictors in a new sample of 219 trauma-affected refugees. The overall aim was to enhance the understanding of factors influencing treatment response in this target group and, ultimately, be able to offer improved treatment.

The replicated predictors for a positive treatment response regarding both PTSD (HTQ) and depression (HAM-D) were age, low pain score (BPI baseline), and all five psychotherapy readiness factors from the CTP predictor index. Furthermore, migrant status was identified as a predictor of PTSD (HTQ), whereas a medium duration of functional impairment and chronic pain were identified as predictors of depression (HAM-D).

### Age, chronicity and migration status replicated as predictors of treatment response

Age was replicated as a predictor of treatment response for both PTSD (HTQ) and depression (HAM-D) patients. For depression (HAM-D), we identified a U-shaped relationship, indicating a greater treatment response, primarily for

**Table 5. Multiple regression results.**

| | | HTQ change pre-post | | | HAM-D change pre-post | | |
|---|---|---|---|---|---|---|---|
| | | β | p-value for β | R² | β | p-value for β | R² |
| **Model 1: All significant ssociodemographic+baseline values** | | | | 0.19 | | | 0.32 |
| Age (c) | | −0.65 | 0.122 | | −1.96 | 0.000*** | |
| Age_squared (c) | | 0.49 | 0.217 | | 1.69 | 0.000*** | |
| Baseline value (c) | | 0.36 | 0.000*** | | 0.37 | 0.000*** | |
| BPI baseline (c) | | −0.22 | 0.009* | | −0.19 | 0.026* | |
| Migrant status refugee (b) | | −0.08 | 0.271 | | −0.01 | 0.904 | |
| Duration of functional impairment: | Medium (b) | −0.07 | 0.286 | | −0.18 | 0.008** | |
| | Long (b) | 0.01 | 0.827 | | 0.04 | 0.403 | |
| **Model 2: Model 1+significant CTP predictor index** | | | | 0.28 | | | 0.39 |
| Age (c) | | −0.72 | 0.074 | | −2.06 | 0.000*** | |
| Age squared (c) | | 0.58 | 0.125 | | 1.84 | 0.000*** | |
| Baseline value (c) | | 0.37 | 0.000*** | | 0.45 | 0.000*** | |
| BPI baseline (c) | | −0.15 | 0.073 | | −0.15 | 0.085 | |
| Migrant status refugee (b | | −0.07 | 0.287 | | 0.00 | 0.964 | |
| Duration of functional impairment: | Medium (b) | −0.06 | 0.309 | | −0.15 | 0.016* | |
| | Long (b) | 0.00 | 0.994 | | 0.04 | 0.454 | |
| Chronic pain (c) | | 0.00 | 0.977 | | 0.11 | 0.136 | |
| Ability to reflect (c) | | 0.01 | 0.958 | | 0.08 | 0.478 | |
| Motivation for active participation (c) | | 0.01 | 0.895 | | 0.05 | 0.678 | |
| Cognitive resources (c) | | 0.07 | 0.522 | | 0.05 | 0.622 | |
| Understanding the concepts of psychotherapy (c) | | 0.17 | 0.130 | | 0.04 | 0.647 | |
| Acceptability of psychological treatment (c) | | 0.11 | 0.280 | | 0.09 | 0.292 | |
| **Model 2b: Alternative model** | | | | 0.28 | | | 0.32 |
| Age (c) | | −0.69 | 0.076 | | −2.07 | 0.000*** | |
| Age squared (c) | | 0.54 | 0.137 | | 1.85 | 0.000*** | |
| Baseline value (c) | | 0.37 | 0.000*** | | 0.45 | 0.000*** | |
| BPI baseline (c) | | −0.16 | 0.058 | | −0.15 | 0.079 | |
| Migrant status refugee (b) | | −0.06 | 0.386 | | 0.00 | 0.985 | |
| Duration of functional impairment: | Medium (b) | −0.06 | 0.293 | | −0.15 | 0.016* | |
| | Long (b) | −0.01 | 0.935 | | 0.03 | 0.495 | |
| Chronic pain (c) | | −0.01 | 0.877 | | 0.11 | 0.144 | |
| Principal component | | 0.30 | 0.000*** | | 0.25 | 0.001*** | |

HTQ: Harvard Trauma Questionnaire. HAM-D: Hamilton Rating Scale Depression. BPI: Brief Pain Inventory.

b: bivariate (1=yes, 2=no). c=continuous.

participants younger than 40 years but also for the oldest group above 60 years. This finding is similar to those of previous studies, where we reported better treatment responses for PTSD and depression among patients aged under 40 years [12,24]. There also seems to be a group of the oldest participants with better treatment responses. However, this latter finding is more uncertain because of the smaller sample size in the older group.

Age was not identified as a predictor in a review from 2020 on factors associated with treatment response from psychological treatments for PTSD, where eight of the 25 included studies were on refugees [45]. Although not found in the review, Resick et al. have identified younger age as a predictor of better treatment response in a study on war veterans with PTSD. Several factors could be related to younger age and have also been found in other populations with PTSD [46]. These factors are related to less stigma attached to accepting mental health treatment [47]; fewer social stressors;

less comorbidity, including somatic comorbidity; and psychotherapy readiness factors, such as cognitive flexibility and openness to engage in psychotherapy [46]. The younger group might have been distressed for a shorter time and had less time since they experienced trauma, which might be associated with fewer chronic conditions that generally respond better to treatment.

In the present study, a medium duration of functional impairment was replicated as a predictor of negative treatment response for depression (HAM-D), likely reflecting the same group of middle-aged patients not responding to treatment [24]. However, we did not find that other factors related to illness chronicity or complexity, such as previous hospitalisation, time since trauma or arrival at the new country [12], or comorbid depression [13], predicted outcomes. This could be explained by the high level of symptoms experienced by a substantial proportion of the participants and the very high frequency of comorbid depression (>95%) [33,34,48].

Migration status, i.e., having arrived in Denmark as a refugee versus being family-reunified with a refugee, was identified as a predictor for positive treatment response [12], However, in the previous predictor study, the identified relation was the opposite – having arrived in Denmark as a refugee was a predictor for a negative treatment outcome [49]. Furthermore, in a previous study focusing specifically on differences in trauma, psychopathology, and treatment response between refugees and family-reunified, no differences were identified in treatment response, and differences were primarily found regarding traumatic exposure [50]. Based on the current analysis, migration status is not identified as a robust predictor of treatment outcome.

## Pain replicated as a predictor of treatment response

In the present study, pain measurements were replicated as predictors of treatment response for both PTSD (BPI) and depressive symptoms (BPI, chronic pain from the CTP predictor index) [24,34]. Those with higher baseline levels of pain improved less during treatment. A study on a similar population of 276 trauma-affected refugees revealed that pain interference (BPI) predicts poorer outcomes for PTSD, depression and anxiety [19]. Research on comorbid mental illness and complaints of pain in refugees is still scarce. There is a need to explore how a focus on relieving pain or the impact of pain can be effectively integrated into the treatment of trauma-affected refugees.

## Factors related to readiness for psychotherapy as predictors of treatment response

The current study points to several factors included in the CTP predictor index assessed by psychologists before the start of psychotherapy as predictors of treatment response across outcomes of PTSD and depression. Although these are subjective evaluations (in contrast to demographic data), the replication shows that these factors include information that seems essential in predicting treatment outcomes. Assessment of readiness for psychotherapy may contribute to identifying patients who are likely to benefit from psychotherapy and may help plan treatment for individual patients [51].

According to the multiple regression model, none of the single psychotherapy readiness factors were significant individual predictors. Nevertheless, the multiple regression analyses revealed an explained variance for the different outcomes of 28--39%, which is rather large in this type of study. However, in the bivariate analyses, they all together contributed to explaining the treatment response variance. In Model 2b, the first principal component of the psychotherapy readiness factors in the CTP predictor index was a highly significant predictor across all five outcome measurements. This finding suggests that we cannot choose one of these factors as more important than the others. There is a need to further understand and explore the content of single factors and their interrelatedness, and further develop and search for robust psychotherapy readiness factors. There is a large amount of research on readiness to change and the importance of motivation, as well as studies on interventions trying to improve treatment response, with interventions focusing on these aspects in other populations [52,53]. Several factors evaluated as part of the CTP predictor index are potentially modifiable [52,54]. However, this field is still unexplored in studies on trauma-affected refugees and requires attention.

## Strengths & limitations

One strength of this study is that the participants in current and previous trials were included in pragmatic randomised trials; thus, the participants are comparable to those referred to similar clinics treating refugees. Another strength is the use of the same data collection methods (variables and outcome measures) across studies. The CTP predictor index developed and utilised in the CTP is not a validated scale. However, in the current study, we assume no specific psychometric properties by using single items and the first principal component as a composite score. Some items in the scale depend highly on a subjective evaluation of qualities that are difficult to assess, such as the ability to reflect. It is, therefore, interesting that although some of the items leave room for individual interpretation of subjective quality, the results regarding associations with outcomes are rather robust. For future studies, when the items included in the index are reviewed, some could be improved to ensure that different clinicians rate them consistently, and continuous use of the index would require regular training to increase interrater reliability. In the current study, all clinicians received standardized training, but no formal interrater reliability checks were performed, and agreement between raters was not systematically monitored. Future studies should include procedures for evaluating and reporting interrater agreement to strengthen the quality of clinician-rated data.

While all tested predictors were theory-driven and based on previous findings, multiple comparisons increase the risk of false positives. Although no formal adjustment was applied, findings should be interpreted with caution, particularly where p-values approach the conventional significance threshold.

## Future research: Possible modifiable factors and factors relevant to clinical practice

In striving to improve services, it is crucial to identify both predictors of patients who benefit from treatment and the barriers to being able to benefit [15]. Among the factors discussed above, potentially modifiable factors are of particular interest in our efforts to improve treatment response. Age is not modifiable per se, but earlier referral could improve treatment response for some patients. The participants' migrant status is not modifiable, but the treatment may be increasingly adapted to specific stressors and needs of daily living. Currently, we are investigating postmigration stressors and cross-sectoral collaboration in an RCT [55].

In the current study, factors related to readiness for psychotherapy could be a target for modification. This should be explored in studies examining the extent to which these factors are modifiable as well as further developing a reliable measure for this concept in trauma-affected refugees.

The group of refugees with PTSD referred to the CTP often have long-lasting severe distress and impaired functioning, and many have been in treatment previously. Thus, the group as a whole is considered challenging to treat. Despite this, we observed statistically significant improvements in mental health and well-being from pre- to posttreatment in the current and previous studies. This finding is rather stable across studies carried out in the past 13 years. These results suggest that further improvement can be obtained by addressing modifiable factors influencing treatment response.

## Conclusions

In conclusion, the findings of this study underscore the importance of identifying and understanding predictors of treatment response in trauma-affected refugees. By replicating and validating previous findings, we have established a clearer picture of the factors that influence treatment efficacy in this vulnerable population.

The findings from our research, highlight key predictors of treatment response that can inform clinical practice and guide future research efforts. First, the identification of modifiable predictors, such as readiness for psychotherapy, suggests that clinicians can enhance treatment outcomes by focusing on these areas. For instance, interventions aimed at increasing motivation and understanding of therapy could be integrated into treatment protocols to improve overall efficacy. Moreover, the recognition of chronic pain as a significant barrier to recovery emphasizes the need for a multidisciplinary approach to treatment. Integrating pain management strategies into mental health interventions may improve

overall treatment efficacy for trauma-affected refugees. This integration could involve collaboration with physical therapists and other healthcare professionals to develop comprehensive care plans that address both psychological and physical health needs.

The U-shaped relationship between age and treatment response also suggests that clinicians should consider age-specific factors when designing treatment plans. Younger patients may benefit from interventions that enhance cognitive flexibility and reduce stigma around mental health treatment, while older patients may require strategies that acknowledge their unique experiences and resilience factors.

As the global refugee crisis continues to escalate, the need for effective mental health treatment strategies becomes increasingly urgent. By focusing on modifiable predictors and integrating multidisciplinary approaches, we can enhance the quality of care for trauma-affected refugees, ultimately leading to improved mental health outcomes and a better quality of life for this vulnerable population. Future research must continue to explore these areas to build upon our understanding and improve the effectiveness of interventions for trauma-affected refugees.

To ensure a more comprehensive and robust evaluation of these predictors, future studies should be conducted across multiple regions and countries, using larger and more diverse samples.

## Supporting information

**S1 Appendix. CTP Predictor index.**
(DOCX)

## Acknowledgments

Special thanks to the participants and staff at the CTP.

## Author contributions

**Conceptualization:** Jessica Carlsson, Hinuga Sandahl.

**Formal analysis:** Jessica Carlsson, Maja Bruhn, Hinuga Sandahl.

**Funding acquisition:** Jessica Carlsson.

**Methodology:** Jessica Carlsson.

**Project administration:** Hinuga Sandahl.

**Writing – original draft:** Jessica Carlsson, Maja Bruhn, Ida Sophie Poschmann, Derrick Silove, Maria Lurenda Westergaard, Erik Lykke Mortensen, Hinuga Sandahl.

**Writing – review & editing:** Jessica Carlsson, Maja Bruhn, Ida Sophie Poschmann, Derrick Silove, Maria Lurenda Westergaard, Erik Lykke Mortensen, Hinuga Sandahl.

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
