## [Decision Letter · Decision Letter 0]

15 Jul 2025

PONE-D-25-20506Identifying robust predictors of treatment response in trauma-affected refugees: results from a randomised controlled trialPLOS ONE

Dear Dr. Sandahl,

Thank you for submitting your manuscript to PLOS ONE. After careful consideration, we feel that it has merit but does not fully meet PLOS ONE’s publication criteria as it currently stands. Therefore, we invite you to submit a revised version of the manuscript that addresses the points raised during the review process.

We look forward to receiving your revised manuscript.

Kind regards,

Mu-Hong Chen, M.D., Ph.D.

Academic Editor

PLOS ONE

Journal Requirements:

[The RCT was supported by grants from TrygFonden (JC, grant number 120354) and Fonden til Lægevidenskabens Fremme (HS, grant number 16-319).]

The funder did not play a role in the study design, data collection or analysis, decision to publish or preparation of the manuscript.].

3. Thank you for stating the following in your manuscript:

[The RCT was supported by grants from TrygFonden and Fonden til Lægevidenskabens Fremme (Foundation for Promotion of Medicine).]

[The RCT was supported by grants from TrygFonden (JC, grant number 120354) and Fonden til Lægevidenskabens Fremme (HS, grant number 16-319).]

4. In the online submission form, you indicated that [The data that support the findings of this study are available upon request from the corresponding author. The data are not publicly available due to privacy and ethical restrictions.].

6. Please include captions for your Supporting Information files at the end of your manuscript, and update any in-text citations to match accordingly. Please see our Supporting Information guidelines for more information: http://journals.plos.org/plosone/s/supporting-information .

Reviewers' comments:

Reviewer's Responses to Questions

**Comments to the Author**

1. Is the manuscript technically sound, and do the data support the conclusions?

Reviewer #1: Partly

Reviewer #2: Yes

Reviewer #3: Yes

2. Has the statistical analysis been performed appropriately and rigorously? 

Reviewer #1: No

Reviewer #2: Yes

Reviewer #3: Yes

3. Have the authors made all data underlying the findings in their manuscript fully available?

Reviewer #1: Yes

Reviewer #2: Yes

Reviewer #3: Yes

4. Is the manuscript presented in an intelligible fashion and written in standard English?

Reviewer #1: Yes

Reviewer #2: No

Reviewer #3: Yes

5. Review Comments to the Author

Reviewer #1: This manuscript presents data analysis from a randomized control trial (RCT) that aims to identify robust predictors of treatment responses, specifically, posttreatment score differences in PTSD symptoms, among trauma-affected refugees in Denmark. The topic is of paramount importance in global public health, given the global burden of refugees. The study was registered as a RCT (with a valid NCT number), and was approved by the respective IRB/Ethics Committee. Some other (statistical) comments were also provided.

1. Methods:

Methods reporting need some work. In the paper, the authors pointed to the reader to look into previous published work on the proposed RCT. However, to maintain the continuity of this current work, some additional effort is needed to briefly reiterate some of the design aspects of the study published in a previous work within this paper.

Specific comments:

(a) For instance, clearly mention the groups that subjects were randomized into. Also, the randomization and allocation concealment should be made very clear (they are NOT the same thing); the trial staff recruiting patients should NOT have the randomization list. Randomization should be prepared by the trial statistician, and he/she would not participate in the recruiting.

(b) Some details on the randomization is needed, which might be already available in the published work. I would advise the authors to put some sentences in that regard with brevity.

(c) Sample size: The sample size/power statement should also be reiterated, with specific reference to the primary outcome measure (posttreatment score differences).

(d) Statistical Analysis Plan:

(d1) A major part of the analysis focuses on regression techniques, which are valid under strong Gaussian assumptions of the errors. Did the authors confirm that Gaussian assumptions when fitting the multiple regression with the response (score differences) were valid? If not, they need to seek alternative methods.

(d2) What's the plan for handling missing values in the design?

2. Results & Conclusions:

(a) The authors should check that any statement of significance should be followed by a p-value in the entire Results section. Otherwise, the Results section look OK; it's pretty straightforward.

(b) Conclusions should state that the current findings are ONLY based on the random samples derived from the refugees entering Denmark. For a comprehensive yet robust evaluation of the predictors, they should allude to future studies involving multiple regions/countries, with much larger sample sizes.

Reviewer #2: This replication study robustly confirms several previously reported predictors of refugee treatment response—especially younger age, lower baseline pain, and five “psychotherapy-readiness” items. Analytic methods are sound. I have some comments for the paper.

Major concerns

1. Methods

Multiple Testing: Discuss risk of false positives and use using p < 0.05 without adjustment.

2. Results

Direction of Effects: Explicitly state whether each predictor is linked to greater or lesser symptom improvement (e.g., refugee status predicted less PTSD change; 5–10 years impairment predicted poorer depression outcome).

3. Results

Table 2 Accuracy: Show denominators for each percentage so figures match actual N. For example, Migrant status / Refugee / 165 / 75.69. N is 218, not 219.

Minor concerns

1. Correct the typos such as “in-depht” to “in-depth” and “1th of June” to “1st of June” in the Methods. Remove stray commas and semicolons as noted (e.g. “we have previously published seven papers identifying different predictors” rather than “predictors of treatment response, (17, …)” which currently has a comma typo).

2. Since the predictor index involves clinician ratings, consider mentioning whether multiple raters were involved and if any training or consistency checks were in place. In the Discussion, you rightly plan to improve inter-rater reliability in future – that’s good. Even a brief mention of current inter-rater agreement (if known) would be useful for readers to gauge the quality of those measures.

3. Use consistent terminology for the CTP predictor index / psychotherapy-readiness construct. For example, if introducing it as the “CTP predictor index” in Methods, later refer to it simply as “predictor index” or psychotherapy readiness index (if you choose that term) rather than varying the name. Similarly, when referring to change in scores, consider stating “improvement (reduction in symptoms)” to remind the reader that a decrease in HTQ/HAM-D is an improvement.

Reviewer #3: Comments: Thank you for giving me an opportunity to review this important study that identifies robust predictors of treatment response in trauma2 affected refugees based on previous randomized studies. Based on the growing number of refugees with mental health problems, the authors are commedable for coming up with this innovation.

Please see below my minor comments.

Can the authors expand on their recommendations to take care of “ none modifiable risk factors”. In the context of refugee settings and cumulative trauma, personal guilt, maladaptive coping behavior of rumination such as the survivors’ persistent thoughts on how justice or revenge against the perpetrators could be achieved perpetuate various mental health comorbidities

6. PLOS authors have the option to publish the peer review history of their article (what does this mean? ). If published, this will include your full peer review and any attached files.

**Do you want your identity to be public for this peer review?** For information about this choice, including consent withdrawal, please see our Privacy Policy .

Reviewer #1: No

Reviewer #2: No

Reviewer #3: No

---

## [Author Response · Author response to Decision Letter 1]

11 Aug 2025

Please find our detailed responses to all reviewer and editor comments in the attached response document. We have addressed each point carefully and revised the manuscript accordingly.

---

## [Decision Letter · Decision Letter 1]

3 Sep 2025

Identifying robust predictors of treatment response in trauma-affected refugees: results from a randomised controlled trial

PONE-D-25-20506R1

Dear Dr. Hinuga Sandahl,

We’re pleased to inform you that your manuscript has been judged scientifically suitable for publication and will be formally accepted for publication once it meets all outstanding technical requirements.

Kind regards,

Mu-Hong Chen, M.D., Ph.D.

Academic Editor

PLOS ONE

Additional Editor Comments (optional):

Reviewer #1:

Reviewer #2:

Reviewers' comments:

Reviewer's Responses to Questions

**Comments to the Author**

1. If the authors have adequately addressed your comments raised in a previous round of review and you feel that this manuscript is now acceptable for publication, you may indicate that here to bypass the “Comments to the Author” section, enter your conflict of interest statement in the “Confidential to Editor” section, and submit your "Accept" recommendation.

Reviewer #1: All comments have been addressed

Reviewer #2: All comments have been addressed

2. Is the manuscript technically sound, and do the data support the conclusions?

Reviewer #1: (No Response)

Reviewer #2: Yes

3. Has the statistical analysis been performed appropriately and rigorously? 

Reviewer #1: (No Response)

Reviewer #2: Yes

4. Have the authors made all data underlying the findings in their manuscript fully available?

Reviewer #1: (No Response)

Reviewer #2: Yes

5. Is the manuscript presented in an intelligible fashion and written in standard English?

Reviewer #1: (No Response)

Reviewer #2: Yes

6. Review Comments to the Author

Reviewer #1: (No Response)

Reviewer #2: (No Response)

7. PLOS authors have the option to publish the peer review history of their article (what does this mean? ). If published, this will include your full peer review and any attached files.

**Do you want your identity to be public for this peer review?** For information about this choice, including consent withdrawal, please see our Privacy Policy .

Reviewer #1: No

Reviewer #2: No

---

## [Editor Report · Acceptance letter]

PONE-D-25-20506R1

PLOS ONE

Dear Dr. Sandahl,

I'm pleased to inform you that your manuscript has been deemed suitable for publication in PLOS ONE. Congratulations! Your manuscript is now being handed over to our production team.

Kind regards,

on behalf of

Dr. Mu-Hong Chen

Academic Editor

PLOS ONE